# From Science to Innovation in Aquatic Animal Nutrition: A Global TRL-Based Assessment of Insect-Derived Feed Technologies via Scientific Publications and Patents

**DOI:** 10.3390/ani15213174

**Published:** 2025-10-31

**Authors:** Cristina M. Quintella, Grace Ferreira Ghesti, Ricardo Salgado, Ana M. A. T. Mata

**Affiliations:** 1Chemistry Institute, Federal University of Bahia, Campus Ondina, Salvador 40170-115, Brazil; 2Center for Energy and Environment, Federal University of Bahia, Campus Ondina, Salvador 40170-115, Brazil; 3MARE—Marine and Environmental Sciences Centre, School of Technology of Setúbal, Polytechnic Institute of Setúbal, Campus Estefanilha, 2910-761 Setúbal, Portugal; ricardo.salgado@estsetubal.ips.pt (R.S.); ana.mata@estsetubal.ips.pt (A.M.A.T.M.); 4Institute of Chemistry, University of Brasília, P.O. Box 4478, Asa Norte, Brasilia 70904-970, Brazil; ghesti.grace@gmail.com; 5LAQV-REQUIMTE, Department of Chemistry, Faculty of Science and Technology, University Nova of Lisbon, 2829-516 Caparica, Portugal

**Keywords:** insect-based feed, aquatic animal nutrition, aquafeed innovation, technology readiness level, TRL, patent analysis, scientific publications, sustainable aquaculture, feed conversion, circular bioeconomy

## Abstract

**Simple Summary:**

Aquaculture plays a critical role in feeding the world but usually requires feed from unsustainable sources like wild fish leftovers. Insects are an alternative because they are nutritious, grow fast, and can be raised on organic waste, turning problems into resources and supporting the circular economy. This study explored how insect-based feeds are being developed and shared worldwide. Unfortunately, despite a significant increase in scientific research, few countries have translated this knowledge into practical technologies. The United States leads in research, and China leads in technological development, but most solutions are used only for local development. Only seven countries export insect-based feeds to others, mainly from the black soldier fly. This shows that more cooperation and investment are needed to make these sustainable feeds available worldwide. Using insects in aquaculture can synergically reduce pressure on oceans, reduce food waste, improve nutrition, create new jobs, and help build a more sustainable food system for people and the planet, presenting themselves more adherent to SDGs 2, 8, 12, 14 and 17.

**Abstract:**

The use of insects for feed has a significant impact on aquaculture, contributing to the achievement of the Sustainable Development Goal of Zero Hunger and Sustainable Agriculture (SDG 2), among others. This study mapped the intermediate Technology Readiness Levels (TRLs), encompassing scientific knowledge (TRL 3) through 971 scientific articles (Scopus) and technological development (TRLs 4–5) through 218 patents (Espacenet). The highest conversions from TRL 3 to TRLs 4–5 were observed for fish, mollusks, crustaceans, and annelids. Key technological targets include carp and black soldier flies (BSF). Most technologies follow circular economy principles. Emerging themes include immunity, cloning, molecular techniques, metabolomics, and genetics. China leads in TRLs 3–5, followed by the United States. Only France, the United States, and five additional countries hold export-oriented patents targeting 26 markets, primarily involving BSF-based feed formulations. Future growth trends are exponential for scientific articles, logarithmic for total patents, and linear for export patents. Collaboration at TRLs 4–5 remains limited, underscoring the need for greater international cooperation to expand access to sustainable insect-based aquaculture feed technologies.

## 1. Introduction

According to the United Nations, within the scope of the 2030 Agenda, achieving Sustainable Development Goal 2—Zero Hunger and Sustainable Agriculture (SDG 2) is essential [1,2]. Indeed, SDG 2 has a strong influence on life expectancy and, consequently, on the Human Development Index, which in turn influences the advancement of several other SDGs. These actions are indispensable for the conservation of our shared home, planet Earth [3,4].

Other SDGs, such as 12 (Responsible Consumption and Production), 11 (Sustainable Cities and Communities), 13 (Climate Action) and 14 (Life Below Water), have been related by waste management, specially agro-industrial residues whose are generated throughout the production process—from raw material collection to final processing—and are distinct from both raw materials and final products. These residues are often discarded without appropriate treatment, resulting in significant socio-environmental impacts [5]. To address issues arising from the improper disposal, increasing attention has been directed toward its reuse and valorization in more sustainable production systems, particularly through the application of insects. This approach seeks to enhance the economic value of industrial processes by converting insects into valuable products, including potential applications in animal and human nutrition [5]. In this context, insect biorefineries have emerged as promising tools, aligning with circular economy principles. The lipid fraction must be removed for nutritional uses, and this component has favorable energy content and composition. Additionally, secondary products such as green grease, biodiesel, and biochar can be derived, serving as alternative energy sources and contributing to the management of organic waste [6].

In 2025, the BRICS—recognized as the largest association of developing countries and a leader among nations with lower Human Development Index scores [7]—unanimously adopted the Joint Declaration with a Focus on Food Security (DCFSA) during the 15th Meeting on Agriculture, Agrarian Development, Family Farming, and Fisheries and Aquaculture [8]. This declaration places strong emphasis on the sustainable management of fisheries and aquaculture (addressed in items 12–15, 17, 21–31, 41, and 42), highlighting their critical role in ensuring food and nutritional security, generating income, sustaining livelihoods, and creating employment opportunities. Furthermore, these sectors are acknowledged for their significant contribution to reducing carbon footprints and strengthening climate resilience, directly aligning with Sustainable Development Goal 13 (Climate Action). The declaration also recognizes their broader positive impact on multiple other Sustainable Development Goals, including SDG 5 (Gender Equality).

The DCFSA highlights the importance of facilitating collaborative partnerships (SDG 17—Partnerships for the Goals) in areas such as governance, best practices, scientific research, innovation, voluntary technology transfer, and capacity building. It also emphasizes alignment with the Food and Agriculture Organization (FAO) of the United Nations and its Blue Transformation Roadmap [9], which seeks to enhance the sustainability and efficiency of aquatic food systems. Additionally, the DCFSA recognizes the value of traditional knowledge in artisanal fisheries, not only as a component of cultural heritage but also as a system of ecosystem stewardship developed over centuries that contributes to employment generation and food and nutritional security.

In 2022, global aquaculture production from captive systems surpassed that of wild capture fisheries, underscoring the sector’s increasing significance [10]. By 2024, global aquaculture production had reached 130 million tons, with feed expenses accounting for approximately 40–60% of total operational costs [11]. Insect-based alternatives address dual challenges: sustainable protein sourcing and organic waste valorization, potentially creating a market valued at USD 2.1 billion by 2030 [12].

Aquaculture plays a critical role in the circular economy and holds considerable potential as a regionally based, sustainable economic model [13,14,15]. Residues from aquaculture can be repurposed across diverse value chains, including animal feed, human consumption, fishmeal production, collagen extraction, biodiesel generation, and fish oil processing. These applications typically require auxiliary technologies such as residue cleaning, cutting, cooking, pressing, and centrifugation [16].

A growing research trend has focused on using insects as feed for aquaculture, particularly due to their nutritional profiles, including fatty acids, proteins, calories, vitamins (e.g., thiamin and riboflavin), and other bioactive compounds [17,18,19,20]. Insect-based feeds also affect gut microbiota and overall fish health, influencing immune and stress responses. Their immunomodulatory effects are highly dependent on the insect species and developmental stage [21]. Additionally, recent studies have examined their digestibility, impact on animal performance, and effects on final product quality [22].

Insect-based feed presents not only promising opportunities for the aquaculture industry but also associated challenges that require targeted strategies, particularly during large-scale implementation [23,24]. It exhibits a significantly lower environmental footprint in terms of land, water, and energy usage [25], positioning it as a viable circular economy solution [26]. Furthermore, its share in aquaculture is expected to grow in the coming years, with several countries already establishing legislation to regulate its use [27].

Several insect species have been investigated as nutritional sources, with emphasis on improving their bioconversion efficiency and enabling the global distribution of insect-based feeds and related products [28,29,30].

The black soldier fly (BSF) (*Hermetia illucens*) has been the most extensively studied species, originally derived from traditional knowledge [31,32,33]. BSFs do the bioconversion of organic waste into animal feed and is widely recognized in the scientific literature as a model organism for this application. Its prominence is attributed to its high efficiency in waste bioconversion, short life cycle, and the favorable nutritional profile of its larvae, which are rich in proteins and lipids [34]. Although BSF remains the most researched and commercially applied insect in this context, other species are also under investigation for similar purposes. These include the mealworm (*Tenebrio molitor*) [35], black cricket (*Gryllus assimilis*) [36], the superworm (*Zophobas morio*) [37], the common housefly (*Musca domestica*) [38]. The use of these insects is actively explored for their potential to replace conventional feed ingredients, such as fishmeal and soy, thereby contributing to more sustainable and circular food production systems. These advantages position BSF as an economically viable solution; however, widespread implementation requires stringent contamination control, regulatory compliance, and process optimization. The technology facilitates circular resource recovery by converting waste into protein and fat for animal feed, with larval composting processes completed in days rather than weeks.

Additionally, studies have reported that BSF bioconversion under optimized conditions has a lower greenhouse gas emission potential. Economic feasibility is further supported by the generation of multiple revenue streams: larvae can be sold whole, dried, or processed into meal and oil, while the resulting frass (excrement) serves as a valuable fertilizer [39,40].

Beyond BSF, silkworm pupae have also been identified as a key insect species used in circular economy applications. These have demonstrated efficacy in enhancing poultry and aquaculture feed formulations by improving growth rates and feed conversion ratios [41]. This integration not only offers a cost-effective protein source but also promotes the sustainable management of sericultural by-products, minimizing waste and improving resource efficiency [42]. With high protein content and palatability, silkworm pupae represent a viable alternative to conventional protein sources such as fishmeal and soybean meal [43]. By converting a sericulture by-product into a valuable feed ingredient, the use of silkworm pupae contributes to addressing environmental concerns by reducing the ecological impact of animal agriculture.

The common housefly (Musca domestica), a ubiquitous insect in daily life, has also been investigated, with studies indicating that its larval meal is microbiologically safe and possesses high nutritional quality [44].

The progression from research to technological development is frequently assessed using Technology Readiness Levels (TRLs) [45,46]. A metric framework can be adopted whereby scientific publications are associated with TRL 3, academic patents with TRL 4, non-academic patents with TRL 5, clinical trials with TRLs 6–8, and patent market potential with TRL 9 [47,48,49].

A comprehensive scientometric analysis mapped TRL 3 by examining articles published between 2013 and 2022, assessing traditional bibliometric indicators and confirming that this field has exhibited steady scientific growth over the past decade [50].

Now the following questions arise: Is the technological development (TRL 4–5) of this emerging field of insects used in aquaculture feed truly growing? What is its relevance? At which TRL stages are these technologies currently positioned? What is their relationship with different types of aquatic organisms? How are technological developments distributed across countries, and which are the target markets? Which are the main organizations leading this research? Are the technologies freely accessible (public domain patents), or are they protected by active patents (granted or pending examination)? Which countries and organizations dominate the market? What are the scientific discoveries and developed technologies, and how are they distributed by TRL?

To answer these questions, articles, total patents, and active patents were mapped globally, along with assignees, countries, organizations, aquatic organism types, technologies, their annual evolution, conversion rates between TRL levels, among other aspects.

## 2. Materials and Methods

To assess TRL 3, the number of scientific articles indexed in the Scopus database was used as a proxy indicator. The search was conducted in April 2025, without any restriction on the publication year, and covered the fields of title, abstract, and keywords. Keywords were applied in Portuguese, English, French, German, and Spanish, following the structure:

*(AQUACULTURE* AND *FEED* AND *INSECTS) AND NOT (PESTS OR PARASITES)*

AQUACULTURE: SUBJECT AREA (AGRI) AND (*aqua?ultur* OR acuicultur*)*FEED: 
*feed* OR aliment* OR nourritur* OR essen**
INSECTS: 
*inse?t* OR lepidoptera* OR hymenoptera* OR hexapod* OR entomolog* OR diptera* OR coleoptera* OR bug* OR beetle* OR arthropoda* OR arthropod* OR cockroach* OR roach* OR periplaneta* OR orthopteran* OR “domestic* inse?t*” OR blattodea* OR blattid* OR blattellidae* OR blattaria* OR barata* OR cucaracha* OR cafard* OR kakerlake**
PESTS OR PARASITES: 
*pest* OR parasit**


This multilingual and taxonomically inclusive search strategy aimed to capture a comprehensive set of scientific literature relevant to the intersection of aquaculture, feed, and insects.

A total of 971 articles up to 2025 were identified, with excellent metadata completeness for titles, abstracts, and publication year (100%), and acceptable coverage for keywords (90.95%) (Appendix A). The data were exported to Excel, saved as a CSV file, and processed in RStudio version 2023.12.1 using Biblioshiny from the Bibliometrix package [51].

To assess TRLs 4–5, the number of patent families—hereafter referred to simply as “patents”—was used as a metric. The patent search was conducted in October 2025 using ORBIT Intelligence by Questel v.2.0.0 software, chosen for its strong capabilities in data processing, statistical reporting, user-friendly interface, and access to the worldwide Espacenet database, which includes data from over 100 countries [49,50]. The same keyword set used for the article search was applied, along with the International Patent Classification (IPC) code A01K61 for “Culture of aquatic animals.” Additionally, the Food Technology domain [52] was included and 218 patents until 2025 were obtained.

All documents were retrieved regardless of publication year or legal status. A second search was performed to identify only active patents, including both granted patents and those pending examination, identifying 78 active patents. The search string used was:

*(A01K-061/IPC* AND *FOOD CHEMISTRY”)/TECT* AND *INSECTS) NOT (PESTS OR PARASITES)*

INSECTS: 
*(INSECT OR LEPIDOPTERA OR HYMENOPTERA OR HEXAPOD OR ENTOMOLOGY OR DIPTERA OR COLEOPTERA OR BUG OR BEETLE OR ARTHROPODA OR ARTHROPOD OR COCKROACH OR ROACH OR PERIPLANETA OR ORTHOPTERAN OR DOMESTIC INSECT OR BLATTODEA OR BLATTID OR BLATTELLIDAE OR BLATTARIA)/TI/AB/OBJ/ADB/ICLM*
PESTS OR PARASITES: 
*PEST* OR PARASIT**


Table 1 presents a brief overview of the patent search scope, summarizing only the most relevant queries. The searches highlighted in gray indicate those selected for in-depth analysis in this study.

Subsequently, the number of documents related to specific aquaculture species was identified in both patent and scientific article datasets, using the search phrases listed in Table 2.

The conversion rate from scientific discovery to technological development (i.e., from TRL 3 to TRLs 4–5) was calculated as the percentage of patents relative to the number of scientific articles, as shown in Equation (1).(1)$${Ratio}_{TRL3\to TRL4-5}=\frac{No.Patents}{No.Articles}$$

The search for active patents was further restricted to those filed in countries other than the first-priority country, i.e., potential patent markets, using the command <NBPC>1> in ORBIT. It was possible to observe that the more recently filled patents were pending or granted (Appendix A).

To evaluate temporal trends, Compound Annual Growth Rates (CAGR) were calculated between triennia, based on the average number of documents per triennium to minimize the influence of atypical years. CAGR was calculated for scientific articles, patents, and active patents. As the latter metric is valid only for 20 years after the patent’s priority date, only the most recent triennia—spaced three years apart—were analyzed. Equation (2) was applied to the periods 2004–2006, 2010–2012, 2016–2018, and 2022–2024.(2)$$CAGR\left(t_{0},\;t_{n}\right)=100\{{\left(\frac{P_{t_{n}}}{P_{t_{0}}}\right)}^{\frac{1}{\left(t_{n}-t_{0}\right)}}-1\}$$
where $t_{0}$ is the start time, $t_{n}$ is the end time, $P_{t_{0}}$ is the number of documents at the start time, $P_{t_{n}}$ is the number of documents at the end time $t_{n}$.

The retrieved data were processed using Microsoft Excel, and composite figures were created with Microsoft PowerPoint.

## 3. Results and Discussion

This section begins by examining the distribution of TRL 3 and TRL 4–5 across different types of aquaculture, considering the taxonomic groups of aquatic organisms targeted by research and technological development. It then presents the annual evolution of scientific articles, patent families in general, and active patents associated with international export potential. The geographical distribution of these developments is analyzed, identifying the leading producing and consuming countries, as well as the organizations holding the largest patent portfolios. The main technological approaches and formulations are described, followed by an assessment of future trends, international collaboration gaps, and key opportunities and challenges for the global dissemination of insect-based aquaculture feeds.

### 3.1. Aquatic Organisms

Initially, the intensity of activity at TRL 3 and TRLs 4–5 was assessed, along with the corresponding conversion rate from scientific discovery to technological development for each type of aquatic organism (Table 3).

As expected, more articles and patents were identified for general aquaculture than for its restriction to feed and insects, consistent with the findings presented in Table 1. Fish, followed by mollusks (which include several types of organisms), exhibit the highest absolute numbers of articles and patents, reflecting humanity’s preferential focus on their aquaculture.

Conversion percentages generally range between 70% and 130%, indicating synchrony between scientific production and technological development. For annelids, patents predominate in general aquaculture, which may be attributed to their challenges of opportunistic feeding behavior. The conversion percentage for fish in general is relatively low for overall aquaculture (27%) and decreases further when insect feed is included (20%), revealing a potential to accelerate technological progress and indicating a bottleneck in technological development. In contrast, crustacean and mollusk technologies show the opposite trend, with higher conversion percentages for insect feed technologies (150%) than for scientific research (107%).

Recent studies (2022–2025) provide strong evidence of technological maturation in insect-based aquaculture feeds, particularly those derived from the BSF. The literature shows clear progression from laboratory formulations to farm-scale trials, with several studies reporting successful high-level fishmeal substitution across multiple species [53,54]. His technological advancement is further evidenced by innovations in substrate tailoring, where larval diets are enriched with fish-processing by-products and hydrolysates to increase the content of EPA/DHA and essential amino acids in the larvae [53].

Additionally, bioprocessing improvements using fermentation of BSF biomass have been applied to enhance digestibility and provide functional benefits tailored to specific species, including improved gut development and effective replacement of marine protein sources [55].

Annelids exhibit high conversion rates both in general aquaculture contexts and in those focused on insect-based feed. These organisms are opportunistic feeders that consume animal carcasses, including insects [56]. However, they represent a significant technological challenge, particularly polychaetes, which bore into shells, leading to the development of various pesticide formulations [57,58,59].

Bivalve mollusks, crustacean mollusks, sponges, sea urchins, and sea cucumbers display higher conversion rates in general aquaculture, indicating that, relative to insect-based feed, these groups remain predominantly within the scientific research phase. This finding is supported by recent species-specific trials showing variable results among aquaculture organisms. For example, studies on Pacific white shrimp (*Litopenaeus vannamei*) have shown that defatted BSF meal can successfully replace fishmeal, enhancing growth performance and flesh quality [60]. Similarly, trials with rainbow trout demonstrated that BSF larvae meal effectively complements high soybean meal diets, improving growth performance, nutritional profiles, and gut health [61].

Shellfish gastropods have incorporated insect meal and grape marc into their diets [62]. Shellfish bivalves are filter-feeding organisms that consume small-sized phyto- and zooplankton, and can potentially ingest fine detritus particles or even sea lice larvae, depending on the taxonomic or size limits [63]. Shellfish crustaceans have also been fed with BSF larvae meal [64].

Even sponges, sea urchins, and sea cucumbers feed on insects, with their feeding habits ranging from predatory to scavenging, suspension-feeding, deposit-feeding, and detritivorous behaviors [65]. Additionally, zooplankton are known predators of Aedes larvae [66,67].

### 3.2. Annual Evolution of Articles, Patents in General and Active Patents for Exportation

From Table 1 and Table 2, it is evident that there is a substantial body of research on aquaculture related to insects and feed; however, is this a recent trend? Figure 1 provides insight into this question.

Patents represent 18% of the total documents, corresponding to a TRL 3 to TRLs 4–5 conversion rate of 36%. Active patents account for only 36% of the total patents (12% pending examination and 24% granted), while 64% are dead (expired) and now in the public domain. At first glance, this may suggest that the technology is approaching maturity.

The cumulative annual evolution of articles, patents, and active patents shows substantial growth since 2014. The apparent stabilization in patent numbers from 2024 onward is an artifact of the 18-month confidentiality period, during which applications are not publicly disclosed, and should therefore be disregarded. A slight stagnation in patent filings during 2021–2022 may be attributed to the effects of the COVID-19 pandemic; however, growth resumes in the post-pandemic period.

When fitting curves to the time-series data to analyze future trends (Appendix A), patents overall exhibit logarithmic growth (81.918 ln(x) + 21.868, R^2^ = 0.9861), which might suggest temporary stagnation. However, scientific development efforts show exponential growth (164.01 exp(0.1605x), R^2^ = 0.9997), and active patents increase linearly (7.7152x − 6.3333, R^2^ = 0.9903), contradicting the general patent pattern that reflects stagnation due to the high proportion of inactive patents (64%). Therefore, this represents a technology with substantial future scientific potential and an emerging technological profile, aligned with global challenges and consistent with bottlenecks identified by the United Nations.

This observation raises several key questions: How is this technology distributed worldwide? Which countries are leading its development? And what are its primary target markets?

### 3.3. Global Distribution of Articles, Patents and Potential Patent Markets

Figure 2 presents world maps offering a global overview of countries active at TRL 3, TRLs 4–5, and TRL 9, based on scientific publications and patent filings (see also Appendix A).

Initially, it is evident that the two world maps displaying the largest number of countries correspond to TRL 3 (Figure 2A) and TRL 9 (Figure 2D), representing published articles and potential patent markets, respectively. This correlation is expected, as the implementation of technology requires personnel with appropriate competencies (Human Readiness Levels) [60]. Scientific production is not only associated with societal capacity building but also enables qualified individuals to conceive and develop new scientific insights, disseminating them globally through publication.

It is also worth noting that the countries with the highest levels of co-authorship in scientific articles are precisely those that file patents.

However, the number of countries that appropriate their technological development knowledge through first-priority patent filings (Figure 2B,C) is considerably lower. This significant reduction—where many countries generate scientific articles and represent prospective markets but do not patent their technological developments—may be attributed to intrinsic limitations in integrating the patent appropriation paradigm within their cultural frameworks for innovation funding and infrastructure support.

Furthermore, when comparing all patents (Figure 2B) with active patents (Figure 2C), some countries identified as potential markets, such as Canada and Australia, are absent. This absence may indicate a reduced or discontinued investment interest in developing aquaculture technologies associated with insects and fish.

It should be pointed out, however, that it is not countries themselves that file first-priority patents, but rather organizations and individuals residing within their geographical boundaries—that is, the actual patent holders.

### 3.4. Patent Holders and Their Technologies

Table 4 lists the ten “parent” organizations holding at least two active patents, along with the number of their active patents and the total number of patents filed.

Additionally, mapping the co-ownership networks associated with the development of these active patents (Figure 3) is essential to understanding the underlying factors that contribute to the maintenance of patent portfolios by these organizations.

The geographic concentration reveals a marked disparity between innovation and commercialization pipelines. While more than 45 countries contribute to research efforts (TRL 3), only 12 countries file patents at TRLs 4–5, with China leading, represented by 7 of the top 10 patent-holding organizations. This concentration may reflect strategic national priorities in food security and aquaculture, favorable regulatory environments for patent protection, or industrial infrastructure that facilitates scaling from research to application. The absence of major aquaculture markets such as Canada and Australia from active patent filings suggests either potential entry points for new technologies or the presence of regulatory or institutional barriers.

Among the organizations analyzed, the Chinese Academy of Fisheries Sciences (CAFS) ranks first in both total patents and active patents. CAFS is a national non-profit scientific research institution, with one-third of its personnel classified as senior researchers. Its research scope includes protection and utilization of fishery resources, aquatic ecology, aquaculture, reproduction, biotechnology, disease control, processing and comprehensive utilization of aquatic products, quality and safety standards, engineering technologies, fisheries equipment, and information systems relevant to the sector [68].

Within CAFS, the Nanjing branch has filed a total of 87 patents. Among the technologies relevant to this study, one active patent addresses ecological reproduction methods for crabs based on pond transformation (CN115589972). The Beijing branch, with 130 inventions filed exclusively in China, focuses on cultivation methods for golden fish (CN109673546). The Heilongjiang River branch, which holds 1082 inventions filed in China and South Africa, patented an artificial river cultivation method for the fish *Glyptosternum maculatum* (CN112772470). The East China Sea branch, with 2260 filed inventions, developed a two-stage cultivation method with enhanced nutrition for the Chinese mitten crab (CN115428755, filed in China and the USA) and constructed ecological facies of algae for indoor blue crab reproduction (CN117502329).

The Anshun branch, located in the southwest of Guizhou Province, has filed a total of 179 inventions. The three patents listed in Table 4 concern the reproduction and feeding of turbellarian worms (CN118542265, CN118556624, CN118844364). Two of these patents have co-ownership (see Figure 3): Guizhou Lvyi Natural Enemy Technology, which has filed five patents in China and the USA, and the Guizhou Research Institute of Chemical Industry through its subsidiary, the Guizhou Institute of Biotechnology, which has filed 77 patents solely in China.

Innovafeed [31] is a biotechnology company that produces insects for animal feed and plant nutrition. Since 2019, it has filed 21 patents in total, including 15 through the European Patent Office (EPO) and 10 via the Patent Cooperation Treaty (PCT). These patents relate to BSF-based feed for shrimp and decapods in general (WO2022/106792, WO2022/106791, WO2022/106790).

The National Institute of Fisheries Science and Development of South Korea holds a total of 8,248 patents filed through Espacenet and in the United States. Patents filed exclusively in South Korea include technologies for the cultivation of Pacific starry flounder using functional feed compositions containing insect-based raw materials (KR10-2373268), continuous flow aquaculture for flounder (KR10-2287178), and moth-based feed compositions for rainbow trout (KR10-2287183). These patents cite Ynsect, a company that holds 40 patents—36 filed through Espacenet and 8 via the PCT.

Anhui Linghang Animal Health Product is a Chinese company that files patents through Shandong Longchang Animal Health Products, which holds a total of 48 patents, all filed exclusively in China. These patents relate to the use of essential oil from *Eucalyptus multicastus* and its microbial inoculum in the preparation of insect repellents for marine fish and crustaceans (CN110537504, CN110583903). Two co-owners are listed: Anhui Yunong Agriculture Technology, with 22 patents filed exclusively in China, and Anhui Fertile Rice Shrimp Farming Specialized Cooperatives, with 17 patents filed exclusively in China.

The Gansu Agricultural Science Institute files patents through the Jiangsu Lixiahe District Agricultural Science Institute, which holds 221 patents filed exclusively in China. The two patents related to this technology concern reproduction methods for large juvenile crayfish and the preparation and use of fermented feed with BSF as the protein source (CN119631947, CN118923775).

Guilin Li River Ecological Science & Technology Development files patents through Liyang Jinquan Ecological Technology Park, which holds 17 patents filed exclusively in China. In relation to the technology discussed herein, its two patents involve high-yield circulating seed cultivation, and methods and devices for the domestication and opening of fingerlings (CN104285873, CN106035173).

Hainan University is a provincial public university located in Haikou, Hainan, China, with a total of 7647 patents filed in the USA and Canada, including 7 through Espacenet and 18 via the PCT. Within the scope of the present technology, its patents concern spawning induction for *Sipunculus australis* (peanut worms) and the large-scale cultivation of *Daphnia suavia* (water fleas) (CN114885870, CN119924233). These patents are co-owned with the Sanya Research Institute, a branch of the same university (Figure 3).

Ynsect [69] is a company operating primarily in machinery, food, and macromolecular chemistry. It began filing its 43 patents in 2014, with 38 applications filed through the European Patent Office (EPO) and 9 via the PCT route. Within the technology field analyzed in this study, the company holds active patents for insect powders used to reduce fish stress (EP3678495) and to prevent fish skeletal deformities (EP3863651).

Shangyu Snake Hot Runner is a newcomer, holding only two patents that refer to breeding insect live bait on water surface and to automatic feeding (CN120202975, CN221554394).

The Sichuan Academy of Agricultural Sciences, an institution with over 470 filed patents mainly focused on specialized machinery, holds patents related to large-scale artificial selective breeding of mandarin fish (CN107114279) and freshwater shrimp (CN105994026). It also co-owns patents with Sichuan Muzhou Technology and Wanyuan Hengkang Agricultural Development (Figure 3).

The Yangling Agricultural Technology Exhibition Center, through the Zhejiang Fisheries Technology Extension Station, has filed 33 patents since 2009, primarily focused on biotechnology, food chemistry, and special machinery. Its patents include innovations related to the domestication of mandarin fish (CN120391359) and blue crab feeding (CN113661950).

It is also noteworthy that assignees from the Republic of Korea and Japan—specifically the National Fisheries Research and Development Institute and Nippon Suisan, respectively—rank among the top organizations in terms of patent portfolio impact (Appendix A).

This predominance of China among the top patent holders with active patents can be attributed to the extensive and geographically distributed aquaculture industry along its coastline, which is periodically monitored by neighboring countries such as Japan to prevent accidental entry of vessels into Chinese waters [70].

It is also important to emphasize that most of these technologies are designed for domestic use, with only 15 patents filed for export purposes—mainly by multinational companies from France (6), the USA (4), Japan (1), Cuba (1), Taiwan (1), Singapore (1), and China (1). A total of 26 countries were identified as potential patent markets. Furthermore, filings were registered in regional patent offices, including PCT (13), EPO (9), and APR (1). The patent holders include Innovafeed, Ynsect, Sumitomo Chemical, Louisiana State University, Aduana General de la República de Cuba, Institut de Recherche pour le Développement (IRD), Timberfish, Alphabet, Rong Shin Jong, Freezem Cryogenics, Flying Spark, Umami Bioworks Pte., and Petidea Capital Investment.

### 3.5. Technologies Description

Figure 4 enables an analysis of the thematic focus of both articles and patents. Additional details can be observed in the technological domains and concept clusters shown in Appendix A, respectively.

It is evident that the thematic scope becomes more restricted as the TRL increases or as patent types become more selective—progressing from articles (Figure 4A), to all patents (Figure 4B), to active patents (Figure 4C), and finally to active patents targeting export markets (Figure 4D). This trend is expected, given that articles and all patents reflect the broader range of knowledge generated, regardless of patent validity or international filings.

Themes related to feed, fish, aquaculture, and insects are consistently present across all categories, aligning with the scope of the search and confirming the adequacy of the keywords and IPC codes adopted.

In the articles, the most recent trends include research on RNA, ribosomes, and 16S rRNA, followed by studies involving topics in immunity, digestion processes, genetics, metabolomics, and biotechnology (Appendix A). The thematic distribution is as follows:Niche themes: genes and bacterial processes;Motor themes: geographic regions and environments, and risks associated with water pollutants and chemical exposure;Emerging themes: cloning and molecular techniques;Basic themes: enterprises and economics, laboratory procedures, genetics, metabolomics, biotechnology, environment, ecology, sustainability, and health.

These themes are further supported by the thematic evolution slices presented in Appendix A.

Topics related to legislation, regulation, and policies become prominent between the periods 2017–2020 and 2021–2025.

Three main thematic clusters are identified (Appendix A):Cluster 1: enterprises and economics;Cluster 2: laboratory procedures;Cluster 3: genetics, metabolomics and biotechnology focusing on reproduction, fertilization, gender, growth, and mortality.

Larvae are present throughout all panels in Figure 4, emphasizing topics such as larval feed, bait, alive larvae patents for fishing (CN221554394), feed formulations for aquatic organisms (WO2022/106791), and commercial distribution to aquaculture farms (EP2265132).

Aquatic plants also emerge as a theme, associated with both pond preparation in shrimp farming (CN117981702) and integrated rice–aquatic animal farming systems (CN108496720, CN109418094, CN109156291). These topics are particularly prominent among active patents, although less frequently found in those targeting export markets.

Among aquaculture organisms, fish, general crustaceans, shrimp, and crabs are the most prominent. Within the insect category, arthropods of the subphylum *Hexapoda* are the most represented.

Carp appears in general patents related to additives and methods for improving aquaculture feed, such as odor control (CN112841088) and live feed cultivation for fingerlings (CN107306861), among other applications.

The BSF is prominent in active patents and patents aimed at export, indicating a clear trend in market direction.

In export-oriented patents, amino acids are associated with the use of insect powder for preventing skeletal deformities in fish and/or enhancing spinal strength during rearing (EP3863651), increasing the attractiveness of insects or their larvae to aquaculture species (EP4464166), and promoting growth in fish and crustaceans (EP1911764).

Insects classified as pests appear in general patents due to the inherent need for aquaculture methods to control them but are absent from both active and export-oriented patents.

General patents also address economic advantages of integrating aquaculture with rice cultivation systems—for example, those involving insects and frogs (CN115669606)—and highlight circular economy concepts applied to agriculture (CN116548377).

The export patents refer to methods to improve the attractiveness of insects or their larvae for aquaculture applications (EP4464166), processes and methods to optimize feed and food production (US10524490), and technologies related to the isolation and cultivation of muscle and adipose cells from crustaceans (EP3911732).

Export patents also cover insect-based reformulated feed for aquatic environments (US20200345039), BSF meal for shrimp (WO2022/106792, WO2022/106791), feed for aquatic animals (EP4166003), formulations to enhance decapod culture performance (WO2022/106790), and growth-stimulating polypeptides for use in fish and crustaceans (EP1911764). Other applications include the use of insect powder to prevent skeletal deformities and/or enhance spinal strength during fish rearing (EP3863651), and methods to reduce or prevent stress in cultured fish (EP3678495).

Regarding larvae, export patents describe neonate BSF larvae with extended shelf life and their production methods (EP4120830), as well as the production of live mini-larvae for feeding aquarium fish, fish fingerlings, and pets (EP2265132).

In terms of circular technologies, the export patents include integrated circular systems for ecological reproduction and planting (US20210219570), and hybrids and cultivars derived from the rice cultivar designated ‘cl151’ (EP2358193).

Thus, the technological primary gaps of insect-feed aquaculture lie not in basic research but in the commercialization, scalability, and technical feasibility required to transform laboratory findings and conceptual developments into products suitable for global markets. It remains necessary to evaluate the capacity to translate this type of broad circular economy concepts into modular and scalable aquaculture systems that can be replicated, tested with alternative waste streams and insect species, and adapted for export to diverse markets and environmental contexts, thus having a real impact in the SDGs.

### 3.6. Future Tendencies and Opportunities

To evaluate future trends, it is essential to determine whether the sector is experiencing acceleration, saturation, or decline. Therefore, temporal tendencies were calculated using the composite indicator CAGR (Figure 5).

For the publication of articles reporting scientific discoveries at TRL 3, the CAGR was positive and stable at about 18% for all periods.

However, to generate impact toward the SDGs, it is essential that maturity progresses to the technological development stage, which is reflected in patent filings. As shown in Figure 5, the CAGR initially increased for both general patents and active patents, reaching rates at about 60% in the intermediated period, being higher than those observed for scientific articles. Nonetheless, a deceleration is evident in the most recent period for general patents, indicating that fewer technologies are being developed and appropriated.

The CAGR for active patents continues to exhibit a positive trend, although at significantly lower rates, suggesting that transnational exporting companies are maintaining their strategic goals.

A particularly concerning observation is the decline in CAGR for general patents, as these typically represent technologies linked to regional development and may have greater potential impact on the SDGs for underprivileged populations—who are less likely to benefit from expensive, imported technologies.

The use of insects in aquaculture feed is clearly expanding, both in terms of scientific discovery and technological development, including the filing of active patents. A substantial portion of technological knowledge (68% of patents) is already in the public domain and therefore available for broader use. Outside China, this percentage is considerably higher, as most active patents are Chinese and valid only within national boundaries.

Among aquatic organisms, technologies predominantly at TRL 4–5 are related to fish and mollusks in general, followed by crustaceans. Within the insect domain, arthropods of the subphylum *Hexapoda* are most prominent, particularly the BSF.

Importantly, the attractiveness of the sector extends beyond innovation and patentability. Life cycle assessments (LCAs) have shown that BSF larvae production can outperform conventional livestock in terms of greenhouse gas emissions per kilogram of protein produced [34,71].

It is important to acknowledge that some LCA studies suggest that, under specific conditions, proteins derived from BSF larvae may produce higher CO_2_ emissions per kilogram than high-yield conventional crops. However, further analyses demonstrate that a substantial portion of the food’s carbon content (41%) is incorporated into larval biomass, whereas a significantly smaller fraction is released into the atmosphere as CO_2_, particularly when compared to slower microbial decomposition processes [34].

Economic assessments indicate that the viability of mid-scale BSFL production facilities may require an initial capital investment ranging from EUR 1 to 2 million [72]. Profitability in such operations is highly dependent on the availability of low-cost or nearly free feed substrates and the ability to diversify revenue through the commercialization of secondary products such as oil, frass (used as fertilizer), and chitin. At present, BSFL-derived products are primarily marketed as premium ingredients for aquaculture and pet food, a positioning that limits production volume and elevates financial risk—conditions further exacerbated by regulatory uncertainties.

For other insect species and broader technological opportunities, the future of the sector hinges on overcoming these challenges and optimizing associated production processes. Although the sector is technically prepared for broader commercialization, progress depends on the harmonization of regulations, expansion of production infrastructure, and establishment of standardized quality metrics. Comprehensive LCAs and technical assessments are critical for identifying carbon and cost-intensive stages (“hotspots”) to guide more integrated and sustainable process development.

Moreover, creating supportive regulatory frameworks, establishing incentives for technology transfer, and promoting international collaboration in the development of technical standards are essential. Targeted interventions—such as direct subsidies or carbon credits for waste valorization—may also play a crucial role in reducing barriers and accelerating sectoral growth.

Indeed, future prospects indicate a strong research foundation with an exponential growth trajectory, alongside clearly identified technology leaders with sustained investment—such as CAFS and Innovafeed—and the presence of companies with demonstrated export potential in specialized applications.

However, several weaknesses require careful consideration. The geographical concentration of innovation represents a potential risk, with 80% of the total patents originating from China, and 65% of the active patents being from France and USA, particularly when combined with the limited multinational collaboration in patent development. Another critical aspect concerns potential regulatory barriers in key markets such as the European Union and North America [73,74].

A strong overlap is observed between the countries producing scientific articles and those identified as target markets for patented technologies. This suggests that such markets possess the qualified personnel required for technological adoption. However, relatively few countries file patents for these technologies, resulting in characteristics akin to monopolistic control and a lack of collaborative patent development.

In general, significant progress is still needed to fulfill the goals of the Joint Declaration on Food Security [8] and advance the FAO Blue Transformation Roadmap [9].

Looking forward, several critical questions remain: Will humanity continue to develop insect-based aquaculture feed technologies through collaborative partnerships that respect and build upon traditional knowledge? Will effective mechanisms for voluntary transfer of science, technology, and capacity building be established and operationalized? Can best practices be genuinely shared to enable sustainable and circular innovation?

Insect-based feed technologies hold considerable promise for contributing to multiple Sustainable Development Goals (SDGs). However, their future impact will depend on measurable outcomes across environmental, economic, and societal indicators.

Looking ahead, the commercialization of insect-based feed technologies appears increasingly viable, provided that the following conditions are met:Harmonization of regulations across major international markets;Expansion and scaling of production infrastructure;Development of standardized quality metrics specific to insect-based feeds).

Integrated multitrophic aquaculture systems—combining fish, mollusks, algae, and crop cultivation (e.g., rice)—are expected to expand due to their enhanced sustainability through synergistic interactions. The integration of circular technologies, particularly those that enable the reuse of waste, is essential.

Given that aquaculture production has now surpassed wild capture fisheries for fish, future decisions by managers, researchers, and institutions will determine whether the development of these technologies is conducted through equitable partnerships that acknowledge and build upon traditional knowledge systems.

It should also be highlighted that the development of the insect industry for aquaculture represents a transformative socioeconomic opportunity, capable of generating numerous local jobs across new production chains—including waste collection, processing, and commercialization—while improving the sector’s trade balance through the substitution of conventional imported ingredients [75].

Insect production, in particular, offers a viable pathway for local and rural communities as well as small producers, who benefit from models requiring low initial investment and characterized by short production cycles. This approach enables income diversification and supports poverty reduction, especially when integrated into circular economy systems that convert local organic waste into high-quality protein. Moreover, the reduced dependence on finite marine resources and the lower carbon and water footprints—compared with soy or fishmeal production—yield significant indirect economic benefits [76].

Nevertheless, for this potential for sustainability, competitiveness, and social inclusion to be fully achieved, investments in large-scale production technologies, the establishment of adequate regulatory frameworks, and the creation of financing and knowledge-transfer programs for small producers remain essential [77].

## 4. Conclusions

Humanity is currently experiencing an era of integrative knowledge, associated with the fifth and sixth technological waves, which combine biotechnology and sustainability with recent advances in information and communication technologies and generative (artificial) intelligence [78]. Technologically, it is noteworthy that significant contributions from advanced genetics and biotechnology to insect-based aquaculture feeds have not yet been reflected in patent filings, suggesting the emergence of a new technological frontier. Furthermore, very few patents incorporate tools related to remote monitoring, generative intelligence, or automation. An additional emerging trend is the development of functional feeds aimed at enhancing the performance of aquatic organisms.

However, for these advancements in insect-based aquaculture feed technologies to materialize, substantial financial investment, infrastructure, and qualified personnel are required. Clear governmental priorities and public policies that support such developments are essential. Moreover, policies must ensure that technology does not become excessively concentrated in a few countries—as is currently evident in the distribution of active patents targeting export markets—in order to avoid technological disparities that may hinder progress toward the SDGs and the 2030 Agenda.

## Figures and Tables

**Figure 1 animals-15-03174-f001:**
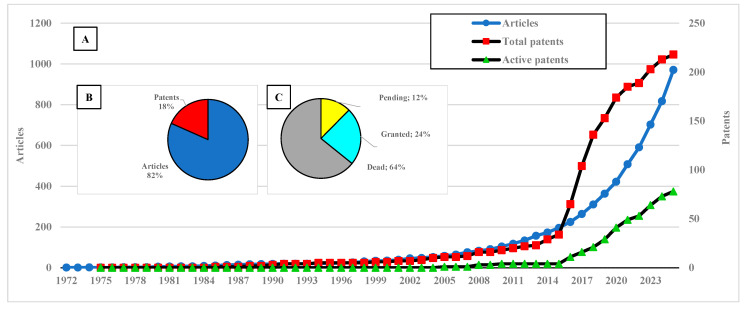
Number of published articles and patents. (**A**) Cumulative annual trends of articles by publication year (left axis, blue circles) and of total patents (right axis, red squares) and active patents (right axis, green triangles) by first priority year. (**B**) Total proportion of articles and patents. (**C**) Distribution of patent legal statuses: dead (expired), pending examination, and granted.

**Figure 2 animals-15-03174-f002:**
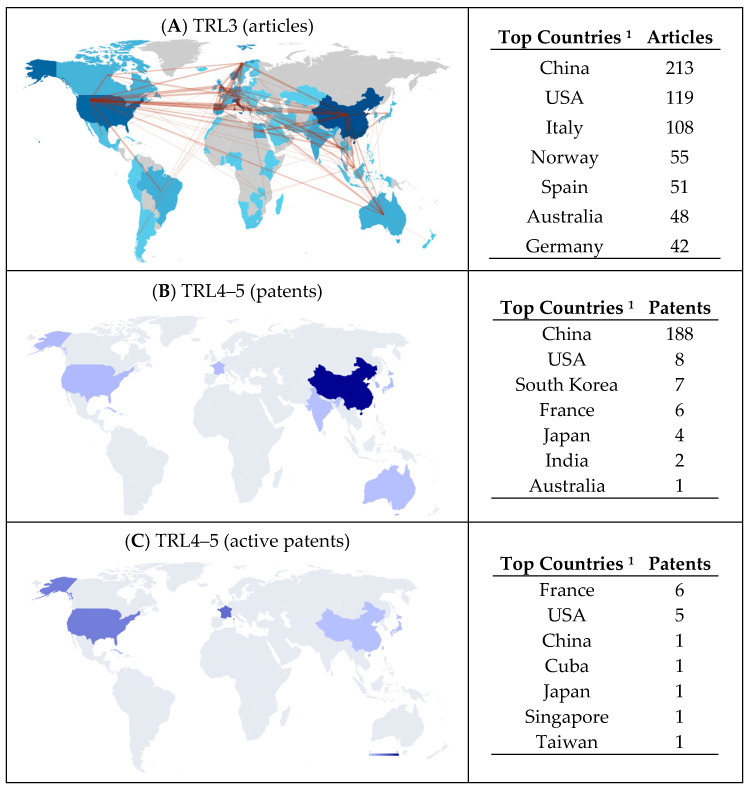
World maps and top-countries tables showing: (**A**) Countries with published scientific articles (TRL 3) and their international collaborations; (**B**) First-priority countries for all patent families (TRLs 4–5); (**C**) First-priority countries for active patents—those pending examination or granted (TRLs 4–5); (**D**) Countries where patents have been filed beyond the first priority, indicating potential commercial markets (TRL 9). ^1^ Data for all countries is available in Appendix A.

**Figure 3 animals-15-03174-f003:**
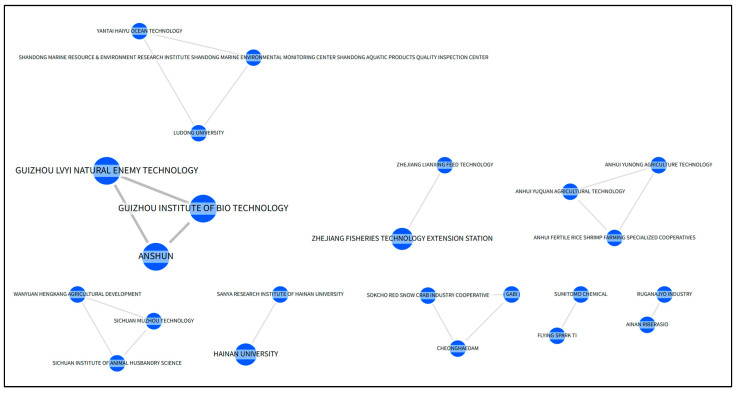
Co-ownership networks of active patents: thick lines for two patents, thin lines for one patent.

**Figure 4 animals-15-03174-f004:**
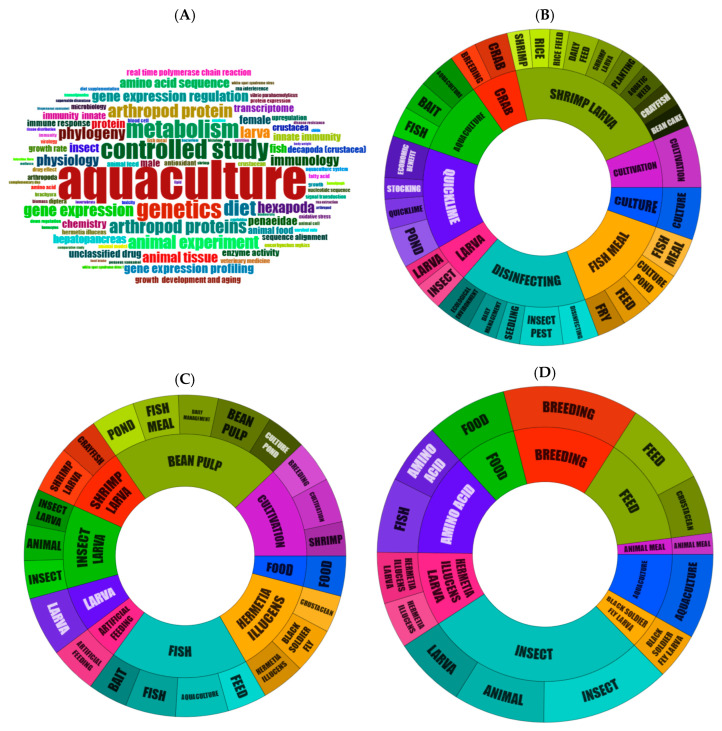
Themes of articles and patents: (**A**) Keyword cloud of articles; (**B**) Diagram of technologies and applications of all patents; (**C**) Diagram of technologies and applications of active patents; (**D**) Diagram of technologies and applications of active patents targeting export.

**Figure 5 animals-15-03174-f005:**
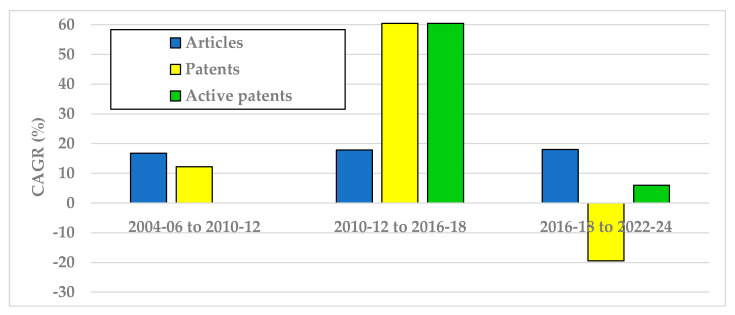
Compound Annual Growth Rate (CAGR) temporal tendencies of articles, general patents, and active patents.

**Table 1 animals-15-03174-t001:** Summary of patent search scope, highlighting in gray the searches analyzed in the present study.

Insects	Aquaculture	Feed	Active Patents	Exportation Patents	No. Patents
	AND				79,268
AND	AND				1297
AND	AND	AND			218
AND	AND	AND	AND		78
AND	AND	AND	AND	AND	15

**Table 2 animals-15-03174-t002:** Keywords used for article searches and International Patent Classification (IPC) codes used for patent searches related to specific aquatic organisms.

Aquatic Organism	IPC Codes	Article Keywords
Bivalve mollusks, e.g., oysters or mussels	A01K61/54	“shellfish bivalve*” OR oyster* OR mussel*
Crustacean mollusks, e.g., lobsters or shrimps	A01K61/59	“shellfish crustacean*” OR lobsters OR shrimps
Gastropod mollusks, e.g., abalones or turban snails	A01K61/51	“shellfish gastropod*” OR abalones OR “turban snails”
Annelids, e.g., lugworms or Eunice	A01K61/40	Annelid* OR lugworm* OR Eunice*
Mollusks in general	A01K61/50	shellfish* OR “shellfish bivalve*” OR oyster* OR mussel* OR “shellfish crustacean*” OR lobsters OR shrimps
Sponges, sea urchins, or sea cucumbers	A01K61/30	sponge* OR “sea urchin*” OR “sea cucumber*”
Fish in general	A01K61/10	fish*

**Table 3 animals-15-03174-t003:** Number of documents for each technology readiness level (TRL) for different types of aquatic organisms, along with their percentage rates: TRL3 (scientific articles) and TRL4–5 (patents).

Aquatic Organism	Aquaculture in General			Insects Feed Aquaculture		
	TRL 3	TRL 4–5	TRL 3 → 4–5 Conversion Rate (%)	TRL 3	TRL 4–5	TRL 3 → 4–5 Conversion Rate (%)
Bivalve mollusks, e.g., oysters or mussels	3469	2480	71%	7	4	57%
Crustacean mollusks, e.g., lobsters or shrimps	5680	6058	107%	44	66	150%
Gastropod mollusks, e.g., abalones or turban snails	673	864	128%	4	9	225%
Annelids, e.g., lugworms or Eunice	131	651	497%	5	6	120%
Mollusks in general	10,290	9548	93%	62	77	124%
Sponges, sea urchins, or sea cucumbers	5962	842	14%	3	2	67%
Fish in general	33,367	8998	27%	394	77	20%

**Table 4 animals-15-03174-t004:** The ten “parent” organizations with at least two active patents, including their number of active patents and total number of patents.

Parent Organization	Country of Origin	Active Patents	Total Patents
CAFS—Chinese Academy of Fisheries Sciences	China	5	9
Anshun Branch	China	3	3
Guizhou Lvyi Natural Enemy Technology	China	3	3
Guizhou Research Institute of Chemical Industry	China	3	3
Innovafeed	France	3	3
Republic of Korea (National Institute of Fisheries Science)	South Korea	3	3
Ynsect	France	2	3
Anhui Linghang Animal Health Product	China	2	2
Gansu Agricultural Science Institute	China	2	2
Guilin Li River Ecological Science & Technology Development	China	2	2
Hainan University	China	2	2
Shangyu Snake Hot Runner	China	2	2
Sichuan Academy of Agricultural Sciences	China	2	2
Yangling Agricultural Technology Exhibition Center	China	2	2

## Data Availability

The original contributions presented in this study are included in the article/Appendix A. Further inquiries can be directed to the corresponding author.

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
