# Peer review of "From Science to Innovation in Aquatic Animal Nutrition: A Global TRL-Based Assessment of Insect-Derived Feed Technologies via Scientific Publications and Patents"

_animals, 2025, doi:10.3390/ani15213174_

Round 1

Reviewer 1 Report

Comments and Suggestions for Authors

1.Insufficient Interpretation of Results:The study provides extensive data and visualizations but lacks in-depth interpretation of the results. For instance, the analysis of the TRL distribution among different aquatic organisms does not explore the underlying reasons for the observed high activity or low conversion rates at specific TRL stages. Factors such as biological characteristics, market demands, and technological challenges, which may influence these outcomes, are not adequately discussed.

Recommendation: The discussion section should be expanded to include a more thorough analysis and interpretation of the results. Integrating relevant knowledge and context from the field to identify potential causes would enhance the depth and value of the study.

2.Unclear Future Trends:The discussion of future trends is vague. While the study mentions potential directions such as accelerated technology maturity and increased patent applications, it lacks specific predictive models and quantitative analyses.

Recommendation: A more definitive forecast of research and technological innovation trends in insect-based feed for aquaculture over the next few years, including confidence intervals or uncertainty analyses, would strengthen the scientific rigor and credibility of the study’s future projections. Methods such as time-series analysis or scenario analysis could be employed for this purpose.

3.Limited Discussion on Socioeconomic Impacts:The study primarily focuses on the technological and scientific aspects of insect-based feed in aquaculture, with minimal attention to its potential socioeconomic impacts. These impacts include job creation, local economic stimulation, and effects on consumer health.

Recommendation: A more comprehensive assessment of the technology’s importance and potential would benefit from an analysis of its broader socioeconomic contributions in the discussion section.

4.Issues with Figure 2:Figure 2, which aims to show the global distribution of countries across different TRL stages, has significant problems. The country boundaries are unclear, labels overlap, and the color distinctions are not pronounced enough. Additionally, the lack of detailed data annotations and a clear legend makes the information hard to interpret.

Recommendation: Consider using a high-resolution map, improving color contrast, annotating country names and data, and refining the legend and explanatory text. Additionally, alternative visualizations such as bar charts or tables could be used to present the data more clearly, making the performance of countries at different TRL stages easier to understand.

Author Response

REVIEWER 1

COMMENTS AND SUGGESTIONS FOR AUTHORS

1.Insufficient Interpretation of Results: The study provides extensive data and visualizations but lacks in-depth interpretation of the results. For instance, the analysis of the TRL distribution among different aquatic organisms does not explore the underlying reasons for the observed high activity or low conversion rates at specific TRL stages. Factors such as biological characteristics, market demands, and technological challenges, which may influence these outcomes, are not adequately discussed.

Recommendation: The discussion section should be expanded to include a more thorough analysis and interpretation of the results. Integrating relevant knowledge and context from the field to identify potential causes would enhance the depth and value of the study.

RESPONSE:

The author thanks the Reviewer. The discussion in Section 3.1 has been expanded. To support the analysis, six additional references have been included, complementing the previous ones (53–61). The text now reads:

Section 3.1  (lines 276-323):

As expected, more articles and patents were identified for general aquaculture than for its restriction to feed and insects, consistent with the findings presented in Table 1. Fish, followed by mollusks (which include several types of organisms), exhibit the highest absolute numbers of articles and patents, reflecting humanity’s preferential focus on their aquaculture.

Conversion percentages generally range between 70% and 130%, indicating synchrony between scientific production and technological development. For annelids, patents predominate in general aquaculture, which may be attributed to their challenges of opportunistic feeding behavior. The conversion percentage for fish in general is relatively low for overall aquaculture (27%) and decreases further when insect feed is included (20%), revealing a potential to accelerate technological progress and indicating a bottleneck in technological development. In contrast, crustacean and mollusk technologies show the opposite trend, with higher conversion percentages for insect feed technologies (150%) than for scientific research (107%).

Recent studies (2022–2025) provide strong evidence of technological maturation in insect-based aquaculture feeds, particularly those derived from the BSF. The literature shows clear progression from laboratory formulations to farm-scale trials, with several studies reporting successful high-level fishmeal substitution across multiple species [53,54]. His technological advancement is further evidenced by innovations in substrate tailoring, where larval diets are enriched with fish-processing by-products and hydrolysates to increase the content of EPA/DHA and essential amino acids in the larvae [53].

Additionally, bioprocessing improvements using fermentation of BSF biomass have been applied to enhance digestibility and provide functional benefits tailored to specific species, including improved gut development and effective replacement of marine protein sources [55].

Annelids exhibit high conversion rates both in general aquaculture contexts and in those focused on insect-based feed. These organisms are opportunistic feeders that consume animal carcasses, including insects [56]. However, they represent a significant technological challenge, particularly polychaetes, which bore into shells, leading to the development of various pesticide formulations [57–59].

Bivalve mollusks, crustacean mollusks, sponges, sea urchins, and sea cucumbers display higher conversion rates in general aquaculture, indicating that, relative to insect-based feed, these groups remain predominantly within the scientific research phase. This finding is supported by recent species-specific trials showing variable results among aquaculture organisms. For example, studies on Pacific white shrimp (Litopenaeus vannamei) have shown that defatted BSF meal can successfully replace fishmeal, enhancing growth performance and flesh quality [60]. Similarly, trials with rainbow trout demonstrated that BSF larvae meal effectively complements high soybean meal diets, improving growth performance, nutritional profiles, and gut health [61].

Shellfish gastropods have incorporated insect meal and grape marc into their diets [62]. Shellfish bivalves are filter-feeding organisms that consume small-sized phyto- and zooplankton, and can potentially ingest fine detritus particles or even sea lice larvae, depending on the taxonomic or size limits [63]. Shellfish crustaceans have also been fed with BSF larvae meal [64].

Even sponges, sea urchins, and sea cucumbers feed on insects, with their feeding habits ranging from predatory to scavenging, suspension-feeding, deposit-feeding, and detritivorous behaviors [65]. Additionally, zooplankton are known predators of Aedes larvae [66,67].

To support this discussion, six additional references have been included:

  1. Bullon, N.; Alfaro, A.C.; Guo, J.; Copedo, J.; Nguyen, T. V.; Seyfoddin, A. Expanding the Menu for New Zealand Farmed Abalone: Dietary Inclusion of Insect Meal and Grape Marc (Effects on Gastrointestinal Mi-crobiome, Digestive Morphology, and Muscle Metabolome). N Z J Mar Freshwater Res 2025, 59, 31–60, doi:10.1080/00288330.2023.2272592.
  2. Webb, J.L.; Vandenbor, J.; Pirie, B.; Robinson, S.M.C.; Cross, S.F.; Jones, S.R.M.; Pearce, C.M. Effects of Temperature, Diet, and Bivalve Size on the Ingestion of Sea Lice (Lepeophtheirus Salmonis) Larvae by Various Filter-Feeding Shellfish. Aquaculture 2013, 406–407, 9–17, doi:10.1016/j.aquaculture.2013.04.010.
  3. Ling, S.-L.Y.; Shafiee, M.; Longworth, Z.; Vatanparast, H.; Tabatabaei, M.; Liew, H.J. Black Soldier Fly Larvae Meal (BSFLM) as an Alternative Protein Source in Sustainable Aquaculture Production: A Scoping Review of Its Comprehensive Impact on Shrimp and Prawn Farming. Anim Feed Sci Technol 2025, 319, 116174, doi:10.1016/j.anifeedsci.2024.116174.
  4. Toma, M.; Bavestrello, G.; Enrichetti, F.; Costa, A.; Angiolillo, M.; Cau, A.; Andaloro, F.; Canese, S.; Greco, S.; Bo, M. Mesophotic and Bathyal Echinoderms of the Italian Seas. Diversity (Basel) 2024, 16, 753, doi:10.3390/d16120753.
  5. Emerson, L.C.; Holmes, C.J.; Cáceres, C.E. Prey Choice by a Freshwater Copepod on Larval Aedes Mosqui-toes in the Presence of Alternative Prey. Journal of Vector Ecology 2021, 46, doi:10.52707/1081-1710-46.2.200.
  6. Russell, M.C.; Qureshi, A.; Wilson, C.G.; Cator, L.J. Size, Not Temperature, Drives Cyclopoid Copepod Pre-dation of Invasive Mosquito Larvae. PLoS One 2021, 16, e0246178, doi:10.1371/journal.pone.0246178.

2.Unclear Future Trends: The discussion of future trends is vague. While the study mentions potential directions such as accelerated technology maturity and increased patent applications, it lacks specific predictive models and quantitative analyses.

Recommendation: A more definitive forecast of research and technological innovation trends in insect-based feed for aquaculture over the next few years, including confidence intervals or uncertainty analyses, would strengthen the scientific rigor and credibility of the study’s future projections. Methods such as time-series analysis or scenario analysis could be employed for this purpose.

RESPONSE:

The author thanks the Reviewer. The time series analysis was performed to predict the future behavior of the series. The text now reads:

Section 3.2 (lines 345-350):

When fitting curves to the time-series data to analyze future trends (Figure S2 in the Supplementary Material), patents overall exhibit logarithmic growth (81.918 ln(x) + 21.868, R² = 0.9861), which might suggest temporary stagnation. However, scientific development efforts show exponential growth (164.01 exp(0.1605x), R² = 0.9997), and active patents increase linearly (7.7152x – 6.3333, R² = 0.9903), contradicting the general patent pattern that reflects stagnation due to the high proportion of inactive patents (64%).

Abstract (lines 44-45):

Future growth trends are exponential for scientific articles, logarithmic for total patents, and linear for export patents.

Supplementary Materials:

Figure S2. Fitted curves for time series used to analyze future trends: (A) Articles; (B) Total pa-tents; (C) Active patents.

3.Limited Discussion on Socioeconomic Impacts: The study primarily focuses on the technological and scientific aspects of insect-based feed in aquaculture, with minimal attention to its potential socioeconomic impacts. These impacts include job creation, local economic stimulation, and effects on consumer health.

Recommendation: A more comprehensive assessment of the technology’s importance and potential would benefit from an analysis of its broader socioeconomic contributions in the discussion section.

RESPONSE:

The authors thank the Reviewer. These impacts and their relevance have been included. The text now reads:

Section 3.1 (lines 698-714):

It should also be highlighted that the development of the insect industry for aquaculture represents a transformative socioeconomic opportunity, capable of generating numerous local jobs across new production chains—including waste collection, processing, and commercialization—while improving the sector’s trade balance through the substitution of conventional imported ingredients [75].

Insect production, in particular, offers a viable pathway for local and rural com-munities as well as small producers, who benefit from models requiring low initial investment and characterized by short production cycles. This approach enables in-come diversification and supports poverty reduction, especially when integrated into circular economy systems that convert local organic waste into high-quality protein. Moreover, the reduced dependence on finite marine resources and the lower carbon and water footprints—compared with soy or fishmeal production—yield significant indirect economic benefits [76].

Nevertheless, for this potential for sustainability, competitiveness, and social inclusion to be fully achieved, investments in large-scale production technologies, the establishment of adequate regulatory frameworks, and the creation of financing and knowledge-transfer programs for small producers remain essential [77].

To support this discussion, three additional references have been included:

  1. Abro, Z.; Macharia, I.; Mulungu, K.; Subramanian, S.; Tanga, C.M.; Kassie, M. The Potential Economic Benefits of Insect-Based Feed in Uganda. Frontiers in Insect Science 2022, 2, doi:10.3389/finsc.2022.968042.
  2. Auzins, A.; Leimane, I.; Reissaar, R.; Brobakk, J.; Sakelaite, I.; Grivins, M.; Zihare, L. Assessing the Socio-Economic Benefits and Costs of Insect Meal as a Fishmeal Substitute in Livestock and Aquaculture. Animals 2024, 14, 1461, doi:10.3390/ani14101461.
  3. Sogari, G.; Bellezza Oddon, S.; Gasco, L.; van Huis, A.; Spranghers, T.; Mancini, S. Review: Recent Advances in Insect-Based Feeds: From Animal Farming to the Acceptance of Consumers and Stakeholders. animal 2023, 17, 100904, doi:10.1016/j.animal.2023.100904.

4.Issues with Figure 2:Figure 2, which aims to show the global distribution of countries across different TRL stages, has significant problems. The country boundaries are unclear, labels overlap, and the color distinctions are not pronounced enough. Additionally, the lack of detailed data annotations and a clear legend makes the information hard to interpret.

Recommendation: Consider using a high-resolution map, improving color contrast, annotating country names and data, and refining the legend and explanatory text. Additionally, alternative visualizations such as bar charts or tables could be used to present the data more clearly, making the performance of countries at different TRL stages easier to understand.

RESPONSE:

The author thanks the Reviewer. The authors included the number of documents for each country. To keep the main text concise, full tables were moved to the supplementary material. World maps were retained for their quick, qualitative, and mnemonic value, each accompanied by a partial table showing the top seven countries.

Section 3.3 (lines 363-367):

Figure 2. World maps and top-countries tables showing: (A) Countries with published scientific articles (TRL 3) and their international collaborations; (B) First-priority countries for all patent families (TRLs 4–5); (C) First-priority countries for active patents—those pending examination or granted (TRLs 4–5); (D) Countries where patents have been filed beyond the first priority, indicating potential commercial markets (TRL 9).

Supplementary Materials:

Table S2(A). Published scientific articles by country (TRL3).

Table S2(B). First-priority countries for all patent families (TRLs 4–5).

Table S2(C). First-priority countries for active patents—those pending examination or granted (TRLs 4–5).

Table S2(D). Countries where patents have been filed beyond the first priority, indicating potential commercial markets (TRL 9).

Reviewer 2 Report

Comments and Suggestions for Authors

This manuscript addresses the application of insects in aquaculture feeds based on published articles and patents. The topic is highly relevant, as it provides insights into the potential of insect-derived resources to alleviate the pressure on conventional feed ingredients and to promote sustainable aquaculture. Nevertheless, the current analysis primarily focuses on the number of publications and patents, without a detailed evaluation of their actual content. This limitation may introduce bias and weaken the reliability of the conclusions. I recommend that the authors carefully screen and analyze only truly relevant studies, so that more robust and scientifically rigorous conclusions can be drawn.

  1. Lines 187–212: While the search strategy is capable of retrieving documents related to “AQUACULTURE AND FEED AND INSECTS”, it is unclear whether all retrieved materials indeed focus on insects as aquaculture feed, rather than merely mentioning insects in passing or in unrelated contexts. For instance, most recent insect-feed studies emphasize fish and crustaceans, yet Table 3 (Line 273) includes a considerable number of references concerning mollusks, sponges, sea urchins, or sea cucumbers. Are these truly studies in which insects were used as aquaculture feeds? I strongly suggest that the authors verify the content of each item, and restrict the analysis to articles and patents that explicitly investigate the use of insects in aquaculture feeds.
  2. Line 317, Figure 2: There seems to be inconsistency in the reported data. Figure 2 indicates that patents account for 23%, whereas the text below states that “Patents represent 21% of the total documents, corresponding to a TRL 3 to TRLs 4–5 conversion rate of 30%.” These numbers do not align. The authors should carefully check the calculation formula (Line 240), and ensure consistency and accuracy across Table 3, Figure 1, and Figure 2. Providing the original data underlying the conversion rates would strengthen the manuscript.
  3. Table 3: Questions also arise concerning the completeness and accuracy of the data. Why are the figures for active patents and total patents inconsistent with the textual descriptions below the table? In addition, the Patent Ranking does not appear in a continuous order. On what basis are certain ranks omitted? Clarification is needed to ensure transparency and reproducibility.

Author Response

REVIEWER 2

COMMENTS AND SUGGESTIONS FOR AUTHORS

  1. This manuscript addresses the application of insects in aquaculture feeds based on published articles and patents. The topic is highly relevant, as it provides insights into the potential of insect-derived resources to alleviate the pressure on conventional feed ingredients and to promote sustainable aquaculture. Nevertheless, the current analysis primarily focuses on the number of publications and patents, without a detailed evaluation of their actual content. This limitation may introduce bias and weaken the reliability of the conclusions. I recommend that the authors carefully screen and analyze only truly relevant studies, so that more robust and scientifically rigorous conclusions can be drawn.

RESPONSE:

The author thanks the Reviewer. Following the reviewer’s suggestions, search scope limitations were added to exclude documents focused on pests or parasites. The root food was removed, while feed was retained to ensure focus on insect feed for aquaculture, with an increased emphasis on aquaculture itself. The analysis was deepened, and new references were included to support the expanded discussion. Each point raised by the reviewer was carefully considered and integrated, improving the overall quality of the manuscript. We now detail each item below.

  1. Lines 187–212: While the search strategy is capable of retrieving documents related to “AQUACULTURE AND FEED AND INSECTS”, it is unclear whether all retrieved materials indeed focus on insects as aquaculture feed, rather than merely mentioning insects in passing or in unrelated contexts. For instance, most recent insect-feed studies emphasize fish and crustaceans, yet Table 3 (Line 273) includes a considerable number of references concerning mollusks, sponges, sea urchins, or sea cucumbers. Are these truly studies in which insects were used as aquaculture feeds? I strongly suggest that the authors verify the content of each item, and restrict the analysis to articles and patents that explicitly investigate the use of insects in aquaculture feeds.

RESPONSE:

The author thanks the Reviewer, whose suggestions improved the quality of this study and manuscript. The search scope was refined by excluding documents related to pesticides or parasites, removing the root food, and restricting the results to the AGRI subject area. Tables, figures, and texts were carefully revised. The texts now read:

  1. Materials and Methods (line 186):

(AQUACULTURE AND FEED AND INSECTS) AND NOT (PESTS OR PARASITES)

  1. Materials and Methods (line 188):

AQUACULTURE: SUBJECT AREA (AGRI) AND (aqua?ultur* OR acuicultur*)

  1. Materials and Methods (lines 217-218):

(A01K-061/IPC AND FOOD CHEMISTRY")/TECT AND INSECTS) NOT (PESTS OR PARASITES)

  1. Materials and Methods (line 195 and line 226):

PESTS OR PARASITES: pest* OR parasit*

Section 3.1 (lines 276-323):

As expected, more articles and patents were identified for general aquaculture than for its restriction to feed and insects, consistent with the findings presented in Table 1. Fish, followed by mollusks (which include several types of organisms), exhibit the highest absolute numbers of articles and patents, reflecting humanity’s preferential focus on their aquaculture.

Conversion percentages generally range between 70% and 130%, indicating synchrony between scientific production and technological development. For annelids, patents predominate in general aquaculture, which may be attributed to their challenges of opportunistic feeding behavior. The conversion percentage for fish in general is relatively low for overall aquaculture (27%) and decreases further when insect feed is included (20%), revealing a potential to accelerate technological progress and indicating a bottleneck in technological development. In contrast, crustacean and mollusk technologies show the opposite trend, with higher conversion percentages for insect feed technologies (150%) than for scientific research (107%).

Recent studies (2022–2025) provide strong evidence of technological maturation in insect-based aquaculture feeds, particularly those derived from the BSF. The literature shows clear progression from laboratory formulations to farm-scale trials, with several studies reporting successful high-level fishmeal substitution across multiple species [53,54]. His technological advancement is further evidenced by innovations in substrate tailoring, where larval diets are enriched with fish-processing by-products and hydrolysates to increase the content of EPA/DHA and essential amino acids in the larvae [53].

Additionally, bioprocessing improvements using fermentation of BSF biomass have been applied to enhance digestibility and provide functional benefits tailored to specific species, including improved gut development and effective replacement of marine protein sources [55].

Annelids exhibit high conversion rates both in general aquaculture contexts and in those focused on insect-based feed. These organisms are opportunistic feeders that consume animal carcasses, including insects [56]. However, they represent a significant technological challenge, particularly polychaetes, which bore into shells, leading to the development of various pesticide formulations [57–59].

Bivalve mollusks, crustacean mollusks, sponges, sea urchins, and sea cucumbers display higher conversion rates in general aquaculture, indicating that, relative to insect-based feed, these groups remain predominantly within the scientific research phase. This finding is supported by recent species-specific trials showing variable results among aquaculture organisms. For example, studies on Pacific white shrimp (Litopenaeus vannamei) have shown that defatted BSF meal can successfully replace fishmeal, enhancing growth performance and flesh quality [60]. Similarly, trials with rainbow trout demonstrated that BSF larvae meal effectively complements high soybean meal diets, improving growth performance, nutritional profiles, and gut health [61].

Shellfish gastropods have incorporated insect meal and grape marc into their diets [62]. Shellfish bivalves are filter-feeding organisms that consume small-sized phyto- and zooplankton, and can potentially ingest fine detritus particles or even sea lice larvae, depending on the taxonomic or size limits [63]. Shellfish crustaceans have also been fed with BSF larvae meal [64].

Even sponges, sea urchins, and sea cucumbers feed on insects, with their feeding habits ranging from predatory to scavenging, suspension-feeding, deposit-feeding, and detritivorous behaviors [65]. Additionally, zooplankton are known predators of Aedes larvae [66,67].

  1. Line 317, Figure 2: There seems to be inconsistency in the reported data. Figure 2 indicates that patents account for 23%, whereas the text below states that “Patents represent 21% of the total documents, corresponding to a TRL 3 to TRLs 4–5 conversion rate of 30%.” These numbers do not align. The authors should carefully check the calculation formula (Line 240), and ensure consistency and accuracy across Table 3, Figure 1, and Figure 2. Providing the original data underlying the conversion rates would strengthen the manuscript.

RESPONSE:

The authors thank the Reviewer for the careful analysis.

Equation 1 (between lines 242 and 243)

Equation 1 had its numerator and denominator inverted and has been corrected.

Section 3.1 (between lines 274 and 275)

The original data underlying the conversion rates are presented in Table 3.

  1. Table 3: Questions also arise concerning the completeness and accuracy of the data. Why are the figures for active patents and total patents inconsistent with the textual descriptions below the table? In addition, the Patent Ranking does not appear in a continuous order. On what basis are certain ranks omitted? Clarification is needed to ensure transparency and reproducibility.

RESPONSE:

The authors thank the Reviewer for the careful analysis. The original Table 3 was renumbered as Table 4, since the manuscript contained two tables labeled as Table 3. Table 3 was revised to improve clarity, and the rightmost column was removed because it was unnecessary for the discussion and caused confusion for readers. Additionally, as the search scope was refined to address the Reviewer’s suggestions, the search month changed from April to October, requiring the table to be updated accordingly. The table was also expanded to include all entities with at least two active patents.

Section 3.4 (lines 391-394):

Table 4 lists the ten “parent” organizations holding at least two active patents, along with the number of their active patents and the total number of patents filed.

Table 4. The ten “parent” organizations with at least two active patents, including their number of active patents and total number of patents.

Section 3.4 (lines 471-488): Four organizations were added, and the text now reads:

Ynsect [69] is a company operating primarily in machinery, food, and macromolecular chemistry. It began filing its 43 patents in 2014, with 38 applications filed through the European Patent Office (EPO) and 9 via the PCT route. Within the technology field analyzed in this study, the company holds active patents for insect powders used to reduce fish stress (EP3678495) and to prevent fish skeletal deformities (EP3863651).

Shangyu Snake Hot Runner is a newcomer, holding only two patents that refer to breeding insect live bait on water surface and to automatic feeding (CN120202975, CN221554394).

The Sichuan Academy of Agricultural Sciences, an institution with over 470 filed patents mainly focused on specialized machinery, holds patents related to large-scale artificial selective breeding of mandarin fish (CN107114279) and freshwater shrimp (CN105994026). It also co-owns patents with Sichuan Muzhou Technology and Wanyuan Hengkang Agricultural Development (Figure 3).

The Yangling Agricultural Technology Exhibition Center, through the Zhejiang Fisheries Technology Extension Station, has filed 33 patents since 2009, primarily focused on biotechnology, food chemistry, and special machinery. Its patents include innovations related to the domestication of mandarin fish (CN120391359) and blue crab feeding (CN113661950).

Round 2

Reviewer 2 Report

Comments and Suggestions for Authors

No recommendations